# OpenReview forum: "Grokking at the Edge of Linear Separability"
_ICLR.cc/2025/Conference — Submitted to ICLR 2025_

### Official Review · Reviewer_tdQU · 2024-11-02

**Soundness:** 2
**Presentation:** 2
**Contribution:** 2
**Rating:** 5
**Confidence:** 3

**Summary:**

This work examines grokking, a phenomenon recently observed in neural network training. The author focused on a simple model for data generation and demonstrated that the occurrence of grokking depends on the ratio between data dimensionality and sample size. This analysis relates to research on implicit bias.

**Strengths:**

This work addresses an interesting and important problem. The analysis relates to key topics in understanding neural networks, such as implicit bias and overfitting.

**Weaknesses:**

1. The paper lacks clear presentation and organization. The connections between the analyses in each section are unclear. Without theorems or propositions in the main body, it is difficult to identify the main conclusions. It is recommended to add a roadmap or flowchart at the end of the introduction to outline the paper's structure, or to include summary statements at the end of each major section to better connect the analyses. Additionally, it is suggested that the authors present key findings as numbered theorems or propositions in the main text to make the primary conclusions more prominent.
2. The logistic loss classification with a single-label model studied in this work is overly simplistic, making it difficult to intuit how the analysis could be extended to other models. It is recommended to discuss potential extensions or limitations of the approach, such as elaborating on its limitations and future work. This could include addressing how the analysis might (or might not) generalize to more complex models.

**Questions:**

1. On page 2, the main contributions are summarized in five points, but only the first point mentions grokking. It’s unclear how the remaining four points relate to grokking. Please consider explicitly connecting each contribution to the central theme of grokking.
2. The single-label model is uncommon; it’s unclear how the analysis of this model extends to others, such as models with binary labels.

---

> ### Author Response · Authors · 2024-11-21
>
> Thank you for reviewing our paper.
> We have addressed your main concerns in detail in the global response. Briefly, we have made the following changes to the manuscript: (a) reorganized the paper to enhance clarity, and (b) generalized our analysis beyond the constant-label approach, demonstrating that Grokking persists near the critical point under broader conditions.
>
> Responses to the questions:
>    1. "It’s unclear how the remaining four points relate to grokking..."
>    We are a bit unsure about the exact concern. All the listed points in the paper are tied directly to mechanisms that enable Grokking in our setup. Our main point is that Grokking is related to criticality, and the remaining points demonstrate why critical behavior comes about in this setting. Explicitly, point 4 discusses "delayed generalization," a concept almost synonymous with Grokking, ubiquitously used in the Grokking literature, for instance in [Humayun et al. 2024, Lee et al. 2024, Liu et al. 2023] and references within.
>
>    2. Over-simplicity of the setup due to constant-labeling:
>    As explained in the global response, we have thoroughly addressed this concern by exploring a more generalized setup. However, the setup itself remains quite simple — which we consider to be a strength rather than a weakness. Simplicity is crucial for developing models that are easy to analyze and that highlight the fundamental elements driving phenomena, which is the essence of the scientific process. In our case, the key takeaway from the setup is that Grokking occurs near critical points in the long-time asymptotic dynamics of training (see the global response for more details).
>
> Regarding your comments, it was hard for us to tease out specific improvements that we could apply. If there are particular aspects you found unclear or challenging to understand, we would greatly appreciate a more detailed criticism.

---

> > ### Comment · Reviewer_tdQU · 2024-11-25
> > **Response to author comment**
> >
> > Thank you for the revisions and explanations. After reviewing the updated version, I’ve raised the score from 3 to 5. I also agree with the other reviewers that the writing and presentation could still benefit from further improvement.

---

> > > ### Author Response · Authors · 2024-11-25
> > > **Improved writing and presentation**
> > >
> > > Dear Reviewer tdQU,
> > >
> > > We would like to bring to your attention our latest revised manuscript, where we have changed much of the text in order to improve readability following your latest comments. In particular, we have highlighted in red the newly phrased theorems, definitions and propositions, as well as the organization of our paper in the introduction. We hope that these changes sufficiently improve our paper in your estimation.
> > >
> > > For all changes, please see the following diff file:
> > > https://filebin.net/vu814u4iv7quxmux
> > >
> > > We kindly ask that if you have any further questions or issues, please let us know, as we would be happy to make any effort to improve our paper further before the end of the discussion period.

---

> > > > ### Author Response · Authors · 2024-12-01
> > > >
> > > > Dear Reviewer tdQU,
> > > >
> > > > Since the discussion period is soon ending, we would greatly appreciate your acknowledgement of our rebuttal and revised manuscript.
> > > >
> > > > We kindly ask that if you have any further questions or issues, please let us know, as we would be happy to answer any additional questions you may have before the end of the discussion period.

---

### Official Review · Reviewer_eABc · 2024-11-03

**Soundness:** 3
**Presentation:** 2
**Contribution:** 1
**Rating:** 5
**Confidence:** 3

**Summary:**

This paper considers a binary logistic regression with isotropic Gaussian inputs and constant (same) labels to study Grokking phenomenon (Power et al., 2022). For the analysis, the authors utilize prior results (Soundry et al., 2018; Nacson et al., 2019; Ji and Telgrasky, 2019) about the learning dynamics of logistic regression. The simplified nature of the setting allows the authors to illustrate Grokking and how it appears in detail. Specifically, they show that Grokking is related to linear separability and whether the data is linearly separable from the origin. Furthermore, they find an "interpolation threshold" for the ratio of input dimension to the number of samples and demonstrate that Grokking occurs near this threshold.

**Strengths:**

- **Studies interesting but simple setting for Grokking**, which is good for understanding the Grokking phenomenon.

- **Provides insights about Grokking:** Specifically, the authors show the relationship between linear separability and Grokking for their setting

- **Easy-to-follow paper:** Although there are some issues (mentioned below) about the writing of the paper, it is mostly easy to read.

- **Detailed examination of the considered setting:** While their setting is oversimplified, the authors investigate Grokking in this setting in a detailed way.

**Weaknesses:**

- **Oversimplified setting:** Due to the oversimplified nature of the setting, it seems like most of the results in this paper are trivial extensions of prior results (Soundry et al., 2018; Nacson et al., 2019; Ji and Telgrasky, 2019) about the learning dynamics of logistic regression. The authors also mention their usage of (or relationship with) these prior results in several places. Therefore, the only additional benefit lies in the particular focus on Grokking. This limits the contribution of this paper. (please see Question 2 below as well)

- **Missing related work about Grokking:** There seems to be missing prior work studying Grokking under more realistic settings (Humayun et al., 2024; Golechha, 2024). Comparison with these works would be beneficial. (please see Question 3 below as well)

- **Wrong jargon usage in the writing**:
  + In the abstract (Line 12), the authors write, "We analyze the asymptotic long-time dynamics of logistic classification on a **random feature model**." Similarly, they mention the "random feature model" in TL;DR as well. However, they study logistic classification using a linear model. By the "random feature model," they meant to say that the input distribution has a Gaussian distribution with isotropic covariance. However, the "random feature model" corresponds to a different model in literature (Rahimi and Rect, 2007).
  + The authors say "the network" when referring to the linear model throughout the paper, which is confusing since there is no network in their setting.
  + The authors write "long times" or "late times" to refer to the "final stages of the training", which is also not common practice in the literature.

- **The introduction does not mention the model and loss function of the setting.** These should be clarified before the contributions are mentioned.

Overall, I think this work should be extended to a more general setting, or at least the authors may show how the insights from the current work translate to examples of more complicated settings, such as more complex data (non-isotropic Gaussian inputs). Furthermore, related work should be extended to include the prior works that are missing and a comparison with them. Finally, the mentioned issues about the writing should be fixed.

(Rahimi and Recht, 2007): Random Features for Large-Scale Kernel Machines (NeurIPS 2007).

(Humayun et. al., 2024): Deep networks always grok and here is why. (ICML 2024)

(Golechha, 2024): Progress Measures for Grokking on Real-world Tasks. (HiLD 2024)

**Questions:**

1. What do the authors mean with $\lesssim$ in Line 215?

2. How do the insights from this work extend to more realistic settings?

3. What is the additional insight of the current work compared to the missing related work (Humayun et al., 2024; Golechha, 2024)?

**Minor Issues**
- The cross-entropy loss in Eq. (1) does not include labels of data, which looks odd at first glance. The authors drop the labels from the equation since all inputs are assigned the same label. However, this should be mentioned as a note near the equation to improve clarity.
- To avoid confusion with empirical accuracy, mentioning "generalization accuracy in Line 137 might be helpful.
- Typo in line 163: "we the present"

**Update:** After the rebuttal, the rating was increased from 3 to 5 since the authors addressed the concerns during the rebuttal.

---

> ### Author Response · Authors · 2024-11-21
>
> We greatly appreciate the time you took to read our manuscript, we found your suggestions very helpful in improving our manuscript.
>    1. Your primary concern seems to be the simplicity of the setup. Indeed, our results are based on Soudry et. al., Nacson et. al, and so on, but our main contribution is not exploring the implicit bias of GD in classification, but rather to show that this bias inevitably leads to a critical point, and that this criticality is responsible for grokking, at least in this scenario (but also in other scenarios). In addition, as you rightfully suggested, we have generalized our results such that the data distribution does not need to be Gaussian, and apply also when the labels are not constant - see the global response for more details.
>
>    2. Regarding the missing related work (and question 3): Thank you for pointing out these works which we were not aware of. They are now cited in the revised manuscript. Our contribution is quite different than both of these works. Golecha et. al. provide ways to measure and quantify progress during gorkking, but do not discuss criticality as the underlying mechanism, which is the main point of our work. Humayun et. al. is more related, but they discuss adversary examples and the geometry of feature space. Their phase-transition is different than ours since it is a dynamic transition (i.e. a change in the regime of training dynamcis as function of time) and not a static one as in our case (i.e. a discontinuity of the $t\to\infty$ limit of generalization). It is a different class of dynamical phenomena.
>
>    3. Regarding other issues (jargon, introduction, and minor concerns): We completely agree. Thanks for pointing these out. We have fixed all these issues in the revised manuscript.
>
>    5. Response to questions:
>       1. We mean $\lambda \rightarrow 0.5^{-}$. We fixed it throughout the paper, thanks.
>       2. Insights for more realistic settings: this is a very important question - we address it in point 2 of the global response.
>       3. We answered above.
>
> If you found our answers satisfactory, we would appreciate reconsidering the score. Regardless, we sincerely thank you again for your detailed comments, which were incredibly helpful.

---

> ### Comment · Reviewer_eABc · 2024-11-22
>
> Thanks for your comment. Could you please highlight the revisions in the paper with a different color so that we can easily spot the changes? Otherwise, maybe you can provide a list of revisions as a comment.

---

> > ### Author Response · Authors · 2024-11-22
> > **Response to reviewer comment**
> >
> > Dear reviewer, we have uploaded a "Differences" pdf file to the following link:
> >
> > https://filebin.net/8rvj1l213cbn51jo
> >
> > Which highlights the changes we have made to the main text.
> > Note that the jargon related comments have been addressed, and the related work section has been updated to include your suggested contributions.
> >
> > Additionally, in the revised manuscript, App. G and App. H are new results for different distributions and discriminating class labels, which should reassure the reviewer that our results are more broad than demonstrated by our original simple example.
> >
> > Lastly, since our model is fully tractable, we are currently working on an appendix which will compare some of the progress measures for grokking apart from weight norm, loss and accuracy, such as adversarial robustness and local complexity.

---

> > > ### Comment · Reviewer_eABc · 2024-11-23
> > >
> > > Thank you for explaining the revisions. After reviewing the authors' responses and changes, I have increased my rating from 3 to 5, as the authors have adequately addressed my initial concerns. However, I believe the writing and presentation could still be improved further, which was also highlighted as a weakness by other reviewers. This is why I have not rated the paper higher.

---

> > > > ### Author Response · Authors · 2024-11-23
> > > >
> > > > We sincerely thank you for your careful assessment of the revised manuscript and for raising our score. In the remaining time, we will make every effort to further enhance the paper's readability and presentation, aligning with the suggestions provided by you and the other reviewers.
> > > >
> > > > If there are any additional specific issues that still trouble you, we would greatly appreciate it if you could point them and we will make sure to address them.

---

> ### Author Response · Authors · 2024-11-25
>
> Dear Reviewer eABc,
>
> We would like to bring to your attention our latest revised manuscript, where we have changed much of the text in order to improve readability following your latest comments. In particular, we have highlighted in red the newly phrased theorems, definitions and propositions. We have also added the loss function to the introduction.
>
> For all changes, please see the following diff file:
> https://filebin.net/vu814u4iv7quxmux
>
> We kindly ask that if you have any further questions or issues, please let us know, as we would be happy to make any effort to improve our paper further before the end of the discussion period.

---

> > ### Author Response · Authors · 2024-12-01
> >
> > Dear Reviewer eABc,
> >
> > Since the discussion period is soon ending, we would greatly appreciate your acknowledgement of our rebuttal and revised manuscript.
> >
> > We kindly ask that if you have any further questions or issues, please let us know, as we would be happy to answer any additional questions you may have before the end of the discussion period.

---

### Official Review · Reviewer_fVQN · 2024-11-05

**Soundness:** 2
**Presentation:** 2
**Contribution:** 2
**Rating:** 6
**Confidence:** 4

**Summary:**

This paper studies a toy model for grokking in binary classification with a linear model trained using gradient flow on gaussian data with constant labels. They argue that grokking occurs when the ratio $d/n$ of parameters to datapoints approaches the critical threshold of ½ — this is the threshold where the data undergoes a sharp transition from being linearly separable to linearly inseparable. Using prior work characterizing gradient flow dynamics for logistic regression, they argue that close when $d/n$ is just barely below ½, it takes a long time to converge to the max-margin separating hyperplane, which explains the sudden shift in test accuracy even though train loss is low.

Edit: After considering the improved writing and new theoretical results in the revision, I have updated my score to a 6.

**Strengths:**

1. It is important to have toy models that we can actually analyze. Even though the model setup is admittedly toy, it still demonstrates interesting train and test time phenomenon.
2. The paper identifies that the margin to linear separability of the data from the origin determines whether the dynamics will grok or not.

**Weaknesses:**

1. I found that the paper was hard to read, and at times imprecise. One suggestion I have is to write the paper in a way where it is easy to identify what concrete properties or statements are being proved.
2. For example, in Line 251, I was originally confused about why can we apply Eq (7), since that result only makes sense for the linearly separable case (especially since it’s talking about hard SVM). It might help if you formally stated the result of Ji and Telgarsky was stated and then also your variant where you include the bias, commenting explicitly on the difference between what “linearly separable” means in these two settings.
3. It seems that the problem setup is (WLOG) assuming that all the labels are -1, so that to classify correctly we should output a negative scalar. Can you explicitly state this (for example, by just defining what the labels are)

**Questions:**

1. Line 197, you should specify that the first 0 is actually the zero vector in $d$ dimensions
2. I know that the Heaviside function $\Theta$ is relatively standard, but could the authors just define it when it is first introduced for readers who might not be familiar. (Minor point, but I feel like it would be more clear anyway if this was just defined as an indicator)
3. Could you change this $\lambda \lesssim \frac{1}{2}$ notation? This might be interpreted as $\lambda$ is bounded within a constant factor of $\frac{1}{2}$. Two candidate suggestions: $\lambda \uparrow \frac{1}{2}$ ot $\lambda \to \frac{1}{2}^-$.
4. The paper cites Wendel’s theorem, which gives an asymptotic 0-1 law for the separability of the training data. Is there a non-asymptotic version of this?
5. Appendix A, line 617. I believe you are missing the word “linearly separable”
6. In Appendix B, it is argued that the parameter $\alpha$ in Power et al. is analogous to that of $\lambda$. However, besides the fact that there is a “phase transition” with respect to these scalar parameters, I don’t see how one can connect them. Could the authors comment on this connection more,

---

> ### Author Response · Authors · 2024-11-21
>
> We wish to thank you for your detailed feedback.
> It seems like your main concern is readability issues with the manuscript. Within the available time-frame, we have made efforts to enhance the quality and presentation of the paper (with plans for further improvements later). Please see the revised manuscript.
>
> Secondly, we have significantly expanded the generality of our model by considering other data-input distribution such as non-isotropic Gaussians and bi-model distributions, as well as non-constant labeling, as detailed in the global response.
>
> Regarding weaknesses 2 and 3: you are correct, thank you for highlighting these weaknesses to us, as they are easily resolved. We have emphasized both points.
> The "constant label" point appears in the abstract and explicitly in the model setup, and the difference between "our" separability and "Soudry/Telgarsky" separability is discussed in Line 195.
>
> Regarding the questions:
>
> 1,2,3, and 5 - Fixed. Thank you.
>
> 4 - Regarding the non-asymptotic version of Wendel's theorem: In Appendix A we provide both the original result of Wendel for finite $N,d$ result (Eq. (17)), which is the CDF of a binomial distribution, and its limiting behavior in the limit $N,d\to\infty$, which is a step function.
>
> 6 - Thank you for raising this important point which requires clarification. The main idea that we want to communicate is that Grokking in both scenarios is related to the fact that the model parameters are in the vicinity of a critical point. In that sense $\lambda$ and $\alpha$ are related as they are the so called "order parameters" that determine criticality. However, there isn't a direct mapping between them, as the two settings likely correspond to a different universality class, see also point 2 in the global response above.
>
> If you found our answers satisfactory, we would appreciate reconsidering the score.
> Regardless, thanks again for the great feedback.

---

> > ### Author Response · Authors · 2024-11-25
> > **Revised manuscript**
> >
> > Dear Reviewer fVQN,
> >
> > We would like to bring to your attention our latest revised manuscript, where we have changed much of the text in order to improve readability. In particular, we have highlighted in red the newly phrased theorems, definitions and propositions. We have also worked hard to make sure the meaning of "separable" in our paper is clarified to mean separable from the **origin**.
> >
> > For all changes, please see the following diff file:
> > https://filebin.net/vu814u4iv7quxmux
> >
> > We kindly ask that if you have any further questions or issues, please let us know, as we would be happy to make any effort to improve our paper further before the end of the discussion period.

---

> > > ### Author Response · Authors · 2024-12-01
> > >
> > > Dear Reviewer fVQN,
> > >
> > > Since the discussion period is soon ending, we would greatly appreciate your acknowledgement of our rebuttal and revised manuscript.
> > >
> > > We kindly ask that if you have any further questions or issues, please let us know, as we would be happy to answer any additional questions you may have before the end of the discussion period.

---

### Official Review · Reviewer_ERfb · 2024-11-07

**Soundness:** 3
**Presentation:** 3
**Contribution:** 2
**Rating:** 8
**Confidence:** 4

**Summary:**

The paper identifies a simple linear classification setting and studies when "grokking" occurs, under Gaussian random data in the linear asymptotic regime. The setup is simplistic, but possibly useful to isolate the emergence of grokking as a result of implicit bias of gradient descent (flow). Relying on analytical tools including simple Gaussian error calculations and advanced implicit bias results from prior works, the authors demonstrate the regime of occurrence of the phenomenon. The results could be of interest to the theoretical statistical learning community, and the insights may be interesting also to the deep learning community.

Update after author response: raising score from 6 to 8

**Strengths:**

S1. The phenomenon of grokking has been of interest as one of the intriguing behaviors of neural network training. As remarked by the authors, this study does not need sophisticated assumptions, unlike some prior work, for demonstrating the phenomenon. The setup is short and the results are clear.

S2. The analytical discussions show a useful application of implicit bias results from literature. While I did not check the derivation in detail, the calculations are sound.

S3. The paper is very well written. The contributions and limitations are clearly discussed.

**Weaknesses:**

To be of greater interest to the current machine learning community, the study may need to address the following weaknesses. That said, in the considered setting, the results are self-contained and interesting on their own.

W1. The impact of the study would be much higher if the analysis can be shown to generalize to more complex models, like discriminative labeling, and non-linear models.

W2. From a practical point of view, having a theoretical model explain an empirical observation is a valuable contribution. In this regard, demonstrating direct counterparts and specific usefulness of the insights obtained from the model to large scale machine learning experiments may be considered. For example, does the "edge of separability" correspond to a certain regime of model size/training in deep models? Can a specific trend in delayed generalization be predicted or explained by the insights obtained?

**Questions:**

Questions:

Q1. The train/test loss and accuracy behaviors are still not fully understood. This paper takes a step in the right direction. While I appreciate simple models yielding interesting insights, from a practical perspective, the study would be much more impactful if the results can be extended to more realistic settings. Some thoughts and questions below:

	a. What are the additional results needed, if one were to try and extend the study to random example classification, but with multiple classes? Perhaps the smallest such extension could be iid normal features and random binary label assignments. From my understanding, there are similar linear separability threshold computations possible (e.g. Kini et al, 2021). Any discussion on this question could be beneficial.
	b. The authors note the study of non-linear regression, motivated by neural network training, to be a feasible extension, with fixed learned last-layer features and a trainable classifier head. My concern here is that it is unclear how the distributions of the last-layer features could be characterized. The features will likely not have nice Gaussian-like properties. Additional modeling of the learned feature extractor may be needed and may complicate the calculations.

Q2. I am trying to reconcile some of the results, and more generally, grokking itself with double descent. For example, one of the statements (line 56) is that in the separable regime, the generalization is worse. On the face, this seems to be different from what double	descent appears to predict. Is the apparent discrepancy stemming from not having any "feature selection" assumption in the model? Does double descent not occur in this linear model? More generally, both the phenomena seem to be connected, in the sense of being a result of the implicit bias of the loss optimization. A discussion elaborating the connection would be helpful to get a better picture of the learning dynamics. Specifically, I am curious about the simplest model that would exhibit both double descent and grokking, which might take the model closer to complex DNNs. A recent work (mentioned by the current paper) Davies et al, 2023 discusses the link between the two.

Q3. The discussion in Section 4 is not fully clear to me. It seems that the authors are constructing a representative deterministic setting in 1D, with the deterministic points constructed to match the macro statistical properties of the random feature model. Such a description could be interesting, with a bit more clarity. For instance, the overparameterization ratio $\lambda$ becomes a parameter of the value of a datapoint. With $d$ being set to 1, $\lambda$ appears to lose its original meaning. Could the authors help explain? The paragraph in App E.1 was somewhat helpful.

---

> ### Author Response · Authors · 2024-11-21
>
> We thank you for your detailed review, and appreciate the positive feedback.
> Regarding the weaknesses:
>    1. We have added to the manuscript an appendix on discriminative labeling and more general input data distributions, please see the global response for more details.
>    2. That's a very good question, we answer it in detail in point 2 of the global response.
>
> Answers to questions:
>
> Q1a. Random labels pose a conceptual problem, since in that case the population distribution is undefined. Could you please clarify? In
> the revised manuscript (appendix H) we provide experiments with binary datasets and show that our results hold qualitatively when the
> dataset is highly imbalanced.
> Q1b. This is a good point. We note, though, that the Gaussianity of the data in this manuscript only serves as a distribution that achieves almost-separable data in some parameter regime. If the dataset is on the verge of separability, we predict that the norm of the over-fitting solution will diverge regardless of the data distribution.
>
> Q2. "double descent" can be used to describe at least two different phenomena: (a) a non-monotonic dependence of the generalization loss on training time (e.g. in Davies et. al. 2024) and (b) a non-monotonic dependence of the asymptotic generalization performance on model complexity (e.g. Schaeffer et. al. ICLR 2024). Our model deals with double-descent of type (a) but not of type (b). Interestingly, both types of double descent are related to critical behavior -- Schaeffer et. al. demonstrate explicitly that the non-monotonicity in the generalization performance is due to an "interpolation threshold" which is a critical point (we alluded to that in lines 55 and 121). Our manuscript, in accord with Davies et. al., show that type (a) is also related.
>
> Q3. Your description is correct. The idea was that the only thing that matters to the dynamics is the distance of the origin from the convex-hull of the dataset (the margin), and in the Gaussian model this is just a function of $\lambda$ in the limit $N,d\to\infty$. Therefore, we replaced the dataset by a simpler dataset consisting of two points only, but with the same dependence of the margin on $\lambda$. We have clarified this in the revised text.

---

> > ### Author Response · Authors · 2024-11-25
> > **Official Comment by Authors**
> >
> > Dear Reviewer ERfb,
> >
> > We would like to bring to your attention our latest revised manuscript, where we have incorporated your requests and comments regarding discriminative labels as well as extended to general data distributions. We have also revised much of the text in the paper to improve readability following suggestions made by the other reviewers, which might make our main points even clearer.
> >
> > We kindly ask that if you have any further questions or issues, please let us know, as we would be happy to make any effort to improve our paper further before the end of the discussion period.

---

> > > ### Author Response · Authors · 2024-12-01
> > >
> > > Dear Reviewer ERfb,
> > >
> > > Since the discussion period is soon ending, we would greatly appreciate your acknowledgement of our rebuttal and revised manuscript.
> > >
> > > We kindly ask that if you have any further questions or issues, please let us know, as we would be happy to answer any additional questions you may have before the end of the discussion period.

---

> > > > ### Comment · Reviewer_ERfb · 2024-12-02
> > > > **Satisfied with response to my questions**
> > > >
> > > > Dear Authors
> > > >
> > > > My sincere apologies for not being able to respond earlier. Thank you for clarifying on some of my questions. Having considered the response and skimmed through the updates, I would like to raise my score to 8. I believe the contributions are useful for extending the theoretical understanding of Grokking. The main weakness remains the gap between theoretical insights and practice. If a rating of 7 was available, that would have been more accurate about my opinion.
> > > >
> > > > Best wishes.

---

### Author Response · Authors · 2024-11-21
**Global response**

We sincerely appreciate the time all of the reviewers dedicated to assessing our paper, and found your comments very helpful in improving its quality.
One concern raised by most reviewers was that the setup appeared overly simplistic, making it unclear how the insights from this model could extend to more complex scenarios.

There are two points to consider here:
   1. The presented model was indeed very simple, but our analysis applies to more general setting. In the revised paper we significantly extended the setup by considering two aspects proposed by the reviewers:

      a) **Other input distribution:**
      Our analytic results hold for **any** input distribution that is symmetric around the origin. The input data needs not have a trivial covariance nor does it need to be Gaussian. In our original presentation we used standard Gaussians as the simplest instructive example, but we realized this was not a good choice. We have now added this discussion to the manuscript, and numerically demonstrate that the results hold for non-isotropic Gaussians as well as bimodal distributions (see Appendix G). Grokking still occurs near $\lambda=0.5$, as this remains a critical point, as discussed in Cover's theorem which holds for points in a "general position".

      b) **Discriminative Labeling Case:**
      In Appendix H, we present a detailed discussion of the discriminative labeling scenario (i.e. not a constant single label for all samples), i.e., binary classification. We show that Grokking can also occur in this case but it is prominent only when the dataset is very unbalanced, close to the "constant labeling" condition. This is not surprising, given that Grokking is known to require fine-tuning of the hyper parameters (as does criticality).


   2. So, what is the bottom line? What can we learn from this simple model, and could the insights be generalized?
      * In our setup, we have shown that Grokking occurs near a critical point in the "long-time dynamics" (final stages of training). Specifically, being close to this critical point ($\lambda=0.5$) creates flat directions in the loss landscape.
      These directions may cause training to stay in the vicinity of almost-stable solutions for arbitrarily long times periods before eventually converging to the global minimum.
      In the physics literature, this behavior is known as "critical slowing down".

      * While we cannot show it rigorously, we conjecture that Grokking is intimately related to such critical points in different settings. In a few examples this has been directly demonstrated (Levi et. al. ICLR 2024, Rubin et. al. ICLR 2024, and to some extent also Liu et. al. ICLR 2023).
      If this is indeed the case, then like in statistical mechanics, there should exist "universality classes" that have similar critical behavior (aka "trends"), but possibly very different underlying mechanisms. This instance of grokking is of a new universality class which was not analyzed in the aforementioned works, and it's signature is the logarithmic divergence of the grokking time as a function of the distance to separability, as is demonstrated in Fig. 1 bottom right and Fig. 4 right.

      * Admittedly, we do not know yet to relate our predictions to concrete examples of Grokking "in the wild". That is, we do not know a system that groks with such logarithmic signature. This may be because grokking examples in the wild belong to a different universality class, or because we do not yet know what to look at. Nevertheless, we think it is insightful to identify grokking as a near-critical phenomenon, and beneficial for the community to analyze different classes. We have edited the manuscript to highlight it.

In summary, we believe that the main strength of this study is that it provides a straightforward model where Grokking occurs, allowing us to identify and analyze its fundamental cause as the proximity to a critical point. In addition, with your valuable feedback and the proposed changes, we believe the scope of the study is significantly broader now.

---

> ### Author Response · Authors · 2024-11-25
>
> Dear reviewers,
>
> We would like to thank you again for engaging with us. We believe your constructive comments have greatly improved our paper.
> Having addressed most of your concerns, it seems that the only common issue that remains is the writing and presentation of the paper.
> We have thoroughly edited the paper to address these concerns, and we believe that the presentation and readability are now significantly improved thanks to your criticism.
> The changes include:
>
>    1.⁠ ⁠Formalizing all claims as numbered propositions, as suggested (marked in red in the updated pdf).
>
>    2.⁠ ⁠Adding a roadmap at the beginning, as suggested.
>
>    3.⁠ ⁠Reorganizing the paper to highlight the main ideas more clearly, making it more accessible to the reader. We have moved some of the
>    discussion regarding the toy model to the appendix and emphasized in the introduction and discussion the main takeaway - that Grokking
>    in this setting results from near-critical behavior, and that this applies also to several other occurrences of grokking in the literature.
>
>    4.⁠ ⁠Addressing additional small issues related to jargon and providing further clarification on points raised by the reviewers. We have
>    highlighted some of the key changes in red, while the rest of the changes are spread throughout the paper.
>
> A diff pdf file (from the initial version) is attached here: https://filebin.net/vu814u4iv7quxmux
>
> We hope that the new revision resolves any remaining concerns regarding the presentation.
> There is not much time left, but if you have any more requests please let us know and we will try our best to address them.

---

### Author Response · Authors · 2024-12-03
**Paper Discussion Phase Summary**

Dear Reviewers, AC, and SAC,

Thank you all again for your time and effort in evaluating our work. We would like to sum up the discussion period from our perspective.
The reviewers raised two main concerns about our manuscript, both have been addressed:

   1. **Generality of our setup and its insights**:
To address this, we demonstrated in the revised manuscript that our model **can** generalize to other input data distributions and discriminative labels. Additionally, we emphasized our conjecture that grokking has a strong link to criticality, which is the main takeaway from our setup. The reviewers acknowledged our response positively and raised the score accordingly.

   2. **Readability of the manuscript**:
To tackle this, we incorporated all reviewer suggestions, including formalizing claims as numbered propositions, addressing jargon, adding a roadmap, and other refinements. This was included in the second revised version.

Currently, it seems the **only remaining concern** relates to readability, as noted by reviewers fVQN, eABc, and tdQU. Reviewer eABc explicitly mentioned that readability was the only reason the score was not higher.

At this point, it appears that reviewers fVQN, eABc, and tdQU have not seen our latest revision yet, as they have not responded. We believe our improvements significantly enhance clarity and hope they will consider a reassessment of the scores during the post-discussion period.

**Update**: Reviewer fVQN has acknowledged the latest revision and raised the score - thanks. This leaves only eABc and tdQU.

Thank you again and best regards,

---

### Public Comment · ~Alon_Beck1 · 2025-07-25
**THIS VERSION IS OUTDATED**

The latest version, accepted to ICML 2025, is available here: https://arxiv.org/abs/2410.04489

---

### Meta-Review · Area_Chair_EbaC · 2024-12-20

**Metareview:**

This paper theoretically investigates the dynamics of binary logistic classification in a toy setup. By identifying the "memorization" and "generalization" solutions to the problem, the paper provides interesting theoretical insights into Grokking dynamics, which have been observed in a broader range of contexts.

While the paper make explicit and interesting connections to Grokking, **all** reviewers find the considered setup overly simplistic, leaving it unclear whether the analysis has broader applicability.
Additionally, Reviewers fVQN, eABc, and tdQU note that the presentation of the paper could be further improved, while Reviewer eABc also points out that the technical contributions of the paper are limited.

This is a borderline paper, and I recommend rejecting it based on the concerns mentioned above.
This was a difficult decision, and I hope the authors can benefit from the discussion and reviewers' comments to further improve the manuscript, particularly in terms of presentation and technical contributions.

**Additional Comments On Reviewer Discussion:**

The reviewers raised the following concerns:
- **All reviewers** find the setup considered in this paper overly simplistic, leaving it unclear whether the analysis is of general interest: The authors interacted with the reviewers during the rebuttal phase to address this concern, but some reviewers (and myself) feel that this issue was **only partially resolved**.
- **Presentation**: Reviewers fVQN, eABc, and tdQU noted that the presentation of the paper could be further improved: While the authors made efforts during the rebuttal phase to address this, this concern was **not fully resolved**.

I have carefully considered all of the above points in making my final decision.

---
Some tiny points:
- please use \citet and \citep appropriately.
- please use ``'' instead of "" in LaTeX (e.g., in the abstract and throughout the manuscript).
- please fix the broken link on Page 21 in the appendix.

---

### Decision · Program_Chairs · 2025-01-22

Reject